# Label Noise-Robust Learning using a Confidence-Based Sieving Strategy

**Reihaneh Torkzadehmahani**  *reihaneh.torkzadehmahani@tum.de*
*Technical University of Munich*

**Reza Nasirigerdeh**  *reza.nasirigerdeh@tum.com*
*Technical University of Munich*
*Helmholtz Munich*

**Daniel Rueckert**  *daniel.rueckert@tum.de*
*Technical University of Munich*
*Imperial College London*

**Georgios Kaissis**  *g.kaissis@tum.de*
*Technical University of Munich*
*Helmholtz Munich*

**Reviewed on OpenReview:** *https://openreview.net/forum?id=3taIQG4C7H*

## Abstract

In learning tasks with label noise, improving model robustness against overfitting is a pivotal challenge because the model eventually memorizes labels, including the noisy ones. Identifying the samples with noisy labels and preventing the model from learning them is a promising approach to address this challenge. When training with noisy labels, the per-class confidence scores of the model, represented by the class probabilities, can be reliable criteria for assessing whether the input label is the true label or the corrupted one. In this work, we exploit this observation and propose a novel discriminator metric called *confidence error* and a *sieving* strategy called CONFES to differentiate between the clean and noisy samples effectively. We provide theoretical guarantees on the probability of error for our proposed metric. Then, we experimentally illustrate the superior performance of our proposed approach compared to recent studies on various settings, such as synthetic and real-world label noise. Moreover, we show CONFES can be combined with other state-of-the-art approaches, such as Co-teaching and DivideMix to further improve model performance*.

## 1 Introduction

The superior performance of deep neural networks (DNNs) in numerous application domains, ranging from medical diagnosis (De Fauw et al., 2018; Liu et al., 2019) to autonomous driving Grigorescu et al. (2020) mainly relies on the availability of large-scale and high-quality data (Sabour et al., 2017; Marcus, 2018). Supervised machine learning in particular requires correctly annotated datasets to train highly accurate DNNs. However, such datasets are rarely available in practice due to labeling errors (leading to *label noise*) stemming from high uncertainty (Beyer et al., 2020) or lack of expertise (Peterson et al., 2019). In medical applications, for instance, there might be a disagreement between the labels assigned by radiology experts and those from the corresponding medical reports (Majkowska et al., 2020; Bernhardt et al., 2022), yielding datasets with noisy labels. Hence, it is indispensable to design and develop robust learning algorithms that

---

*The code is available at: **https://github.com/reihaneh-torkzadehmahani/confes**

are able to alleviate the adverse impact of noisy labels during training. Throughout this paper, we will refer to these methods as *label noise learning* methods.

In the literature, there are different types of label noise including symmetric noise, pairflip noise, and instance-dependent noise. In *symmetric noise* a sample is allocated a random label, while in *pairflip noise* the label of a sample is flipped into the adjacent label (Patrini et al., 2017; Xia et al., 2020; Bai et al., 2021). In real-world scenarios, a corrupted label assigned to a sample depends on the feature values and the true label of the sample, known as *instance-dependent noise* (Liu, 2021; Zhang et al., 2021b). Training DNNs in the presence of label noise can lead to memorization of noisy labels and consequently, reduction in model generalizability (Zhang et al., 2021a; Chen et al., 2021b).

Some of the existing studies for dealing with label noise focus on learning the *noise distribution*. Patrini et al. (2017); Berthon et al. (2021); Yao et al. (2021); Xia et al. (2019); Yao et al. (2020) model the noise distribution as a transition matrix, encapsulating the probability of clean labels being flipped into noisy ones and leverage loss correction to attenuate the effect of the noisy samples.

Other studies (Cheng et al., 2020; Xia et al., 2021; Wei et al., 2020) learn the *clean label distribution* and capitalize on regularization or selection of reliable samples to cope with the noisy labels. A main challenge in this line of work, also known as *sample sieving* (or *sample selection*), is to find a reliable criterion (or metric) that can efficiently differentiate between clean and noisy samples. The majority of the previous studies (Jiang et al., 2018; Han et al., 2018; Yu et al., 2019) employ the *loss value* to this end, where the samples with small loss values are considered to likely be clean ones (*small-loss trick*). A prior study (Zheng et al., 2020) proposes a confidence-based criterion and shows the label is likely noisy if the model confidence in that label is low. Our work lies in this category of confidence-based sieving metrics.

Since learning noise distributions is challenging, it is rarely used in practice. Sample sieving methods, on the other hand, have multiple unsolved problems: They might not always be capable of effectively filtering out noisy labels without supplementary assistance (e.g., additional model in Co-teaching). Moreover, their performance might not be satisfactory in the presence of certain types of noises such as instance-dependent or higher levels of noise. This motivates us to develop new metrics and learning algorithms that are more robust against various types and levels of label noise with minimal additional computational overhead.

**Contributions.** Our main contributions can be summarized as follows:

- We introduce a novel metric called *confidence error* to efficiently discriminate between clean and noisy labels. The confidence error metric is defined as the difference between the softmax outputs/logits of the predicted and original label of a sample. Moreover, we provide a theoretical bound on the probability of error for the proposed metric. Our theoretical analysis and observations indicate there exists a clear correlation between the confidence error value and the probability of being clean. That is, a sample with a lower confidence error has a much higher probability to be a clean sample than a noisy one.

- We then integrate the confidence error criterion into a learning algorithm called *CONFidence Error Sieving* (CONFES) to robustly train DNNs in the instance-dependent, symmetric, and pairflip label noise settings. The CONFES algorithm computes the confidence error associated with training samples at the beginning of each epoch and only incorporates a subset of training samples with lower confidence error values during training (i.e., likely clean samples).

- We validate our findings experimentally showing that CONFES significantly outperforms the state-of-the-art learning algorithms in terms of accuracy on typical benchmark datasets for label noise learning including CIFAR-10/100 (Krizhevsky et al., 2009) and Clothing1M (Xiao et al., 2015). The superiority of CONFES becomes particularly pronounced in scenarios where the noise level is high or when dealing with more intricate forms of noise such as instance-dependent noise.

- We moreover demonstrate that combining CONFES with other learning algorithms including Co-teaching (Han et al., 2018), JoCor (Wei et al., 2020), and DivideMix (Li et al., 2020) provides further accuracy gain, illustrating synergy between CONFES and the existing research endeavors in the field of learning with label noise.

## 2 Related Work

Overcoming the memorization of noisy labels plays a crucial role in label noise learning and improves model generalization by making the training process more robust to label noise (Zhang et al., 2021a; Arpit et al., 2017; Natarajan et al., 2013). The research community mainly tackled the memorization problem by adjusting the loss function (known as *loss correction*), using *implicit/explicit regularization* techniques, or refining the training data and performing *sample sieving.*

Adjusting the loss function according to the noise transition probabilities is an effective method for decreasing the adverse impact of noisy samples during the training but comes at the cost of accurate estimation of the transition matrix (Patrini et al., 2017). Previous studies have paved the way for this non-trivial estimation in different ways. For instance, T-Revision (Xia et al., 2019) estimates the transition matrix without requiring anchor points (the data points whose associated class is known almost surely), which play an important role in the effective learning of the transition matrix. Dual-T (Yao et al., 2020) first divides the transition matrix into two matrices that are easier to estimate and then aggregates their outputs for a more accurate estimation of the original transition matrix.

Another line of work improves model generalization by introducing regularization effects suitable for learning with noisy labels. The regularization effect may be injected *implicitly* using methods such as data augmentation and inducing stochasticity. For example, *Mixup* (Zhang et al., 2018) augments the training data using a convex combination of a pair of examples and the corresponding labels to encourage the model to learn a simple interpolation between the samples. SLN (Stochastic Label Noise) (Chen et al., 2021a) introduces a controllable noise to help the optimizer skip sharp minima in the optimization landscape.

Although the implicit regularization techniques have been proven effective in alleviating overfitting and improving generalization, they are insufficient to tackle the label noise challenge (Song et al., 2022). Thus, the community came up with *explicit* regularization approaches such as ELR (Early-Learning Regularization) (Liu et al., 2020) and CDR (Xia et al., 2021). ELR is based on the observation that at the beginning of training, there is an early-learning phase in which the model learns the clean samples without overfitting the noisy ones. Given that, ELR adds a regularization term to the Cross-Entropy (CE) loss, leading the model output toward its own (correct) predictions at the early-learning phase. Similarly, CDR first groups the model parameters into critical and non-critical in terms of their importance for generalization and then penalizes the non-critical parameters.

A completely different line of work is sample sieving/selection, which aims to differentiate the clean samples from the noisy ones and employ only the clean samples in the training process. The previous works in this direction exploit loss-based or confidence-based metrics as the sample sieving criteria. MentorNet (Jiang et al., 2018) uses an extra pre-trained model (mentor) to help the main model (student) by providing it with small-loss samples. The decoupling algorithm (Malach & Shalev-Shwartz, 2017) trains two networks simultaneously using the samples on which the models disagree about the predicted label. Co-teaching (Han et al., 2018) cross-trains two models such that each of them leverages the samples with small-loss values according to the other model. Co-teaching+ (Yu et al., 2019) improves Co-teaching by considering clean samples as those that not only have small loss but also those on which the models disagree. JoCoR (Wei et al., 2020) first computes a joint-loss to make the outputs of the two models become closer, and then it considers the samples with small loss as clean samples. The utilization of two models in MentorNet, decoupling, Co-teaching, Co-teaching+, and JoCoR leads to a computational inefficiency that is twice as high compared to traditional training methods. LRT (Zheng et al., 2020) employs the likelihood ratio between the model confidence in the original label and its own predicted label and then selects the samples according to their likelihood ratio values. Our work is closely related to LRT as both employ a confidence-based metric for sample sieving. However, our metric is an absolute metric that captures the difference between the model's confidence in the given label and the predicted label. In contrast, LRT is a relative metric that is more sensitive to the model's quality.

Our study belongs to the category of sample selection methods and capitalizes on model confidence to discriminate between the clean and noisy samples akin to LRT, without the need for training an additional model as required by methods like Co-teaching.

# 3 CONFES: CONFidence Error based Sieving

We first provide a brief background on the training process for the classification task. Then, we introduce the proposed confidence error metric and provide a theoretical analysis of its probability of error. Afterward, we present the CONFES algorithm, which capitalizes on confidence error for effective sample sieving.

## 3.1 Background

We assume a classification task on a training dataset $D = \{(x_i, y_i) \mid x_i \in X, y_i \in Y\}_{i=1}^n$, where $n$ is the number of samples and $X$ and $Y$ are the feature and label (class) space, respectively. The neural network model $\mathcal{F}(X_b; \theta) \in \mathbb{R}^{m \times k}$ is a $k$-class classifier with trainable parameters $\theta$ that takes mini-batches $X_b$ of size $m$ as input. In real life, a sample might be assigned the wrong label (e.g., due to human error). Consequently, *clean* (noise-free label) training datasets might not be available in practice. Given that, we assume $\tilde{Y} = \{\tilde{y}_i\}_{i=1}^n$ and $\tilde{D} = \{(x_i, \tilde{y}_i)\}_{i=1}^n$ indicate the noisy labels and noisy dataset, respectively. The training process is conducted by minimizing the empirical loss (e.g., cross-entropy) using mini-batches of samples from the noisy dataset:

$$\min_{\theta} \mathcal{L}(\mathcal{F}(X_b; \theta); \tilde{Y}_b) = \min_{\theta} \frac{1}{m} \sum_{i=1}^m \mathcal{L}(\mathcal{F}(x_i; \theta), \tilde{y}_i), \tag{1}$$

where $\mathcal{L}$ is the loss function and $(X_b, \tilde{Y}_b)$ is a mini-batch of samples with size $m$ from the noisy dataset $\tilde{D}$. Table 1 provides a comprehensive summary of all the notations used in the theoretical analysis, along with their respective definitions.

Table 1: Summary of notations and their definitions

| Notation | Defenition | Notation | Defenition |
|---|---|---|---|
| $(x_i, \tilde{y}_i)$ | Sample $i$ with features $x_i$ and possibly noisy label $\tilde{y}_i$ | $y_i'$ | Predicted label for a sample $i$ |
| $n$ | Total number of samples | $k$ | Total number of classes/labels |
| $\mathcal{F}(\cdot; \theta)$ | A classifier with weights $\theta$ | $H^*(\cdot)$ | The optimal Bayes classifier |
| $\sigma(\cdot)$ | Softmax activation function | $C^{(l)}$ | Model confidence for label $l$ |
| $\mathcal{P}_j(\cdot)$ | True conditional probability for label $j$ | $\tilde{\mathcal{P}}_j(\cdot)$ | Noisy conditional probability for label $j$ |
| $\mathcal{L}(\cdot)$ | Loss function (e.g., cross-entropy) | $\tau_{lj}$ | The probability that label $l$ is flipped to label $j$ |
| $v$ | The best prediction of Baye's optimal classifier | $w$ | The second-best prediction of Baye's optimal classifier |
| $\alpha$ | Sieving threshold | $E_C(\cdot)$ | Confidence error for a sample |
| $\epsilon$ | Maximum approximation error of the classifier | $\psi$ | A placeholder variable |
| $\mathcal{O}$ | Order/asymptotic notation (the upper bound of complexity) | $\mu, \beta, \gamma$ | Tsybakov noise condition variables($\mu \in (0,1]$, $\beta, \gamma > 0$) |

In the presence of label noise, the efficiency of the training process mainly depends on the capability of the model to distinguish between clean and noisy labels and to diminish the impact of noisy ones on the training process. In this study, we propose an elegant metric called *confidence error* for efficient sieving of the samples during training.

## 3.2 Confidence Error as the Sieving Metric

Consider a sample $s = (x_i, \tilde{y}_i)$ from the noisy dataset $\tilde{D}$. The $k$-class/label classifier $\mathcal{F}(x_i; \theta)$ takes $x_i$ as input and computes the weight value associated with each class as output. Moreover, assume $\sigma(\cdot)$ is the softmax activation function such that $\sigma(\mathcal{F}(x_i; \theta)) \in [0,1]^k$ takes classifier's output and computes the predicted probability for each class. We define the *model confidence* for a given label $l \in \{1, \ldots, k\}$ associated with sample $s$ as the prediction probability assigned to the label:

$$C^{(l)} = \sigma(\mathcal{F}(x_i; \theta))^{(l)} \tag{2}$$

The class with the maximum probability is considered as the predicted class, i.e. $y_i'$, for sample $s$:

$$y_i' = \arg\max_{j \in \{1, \ldots, k\}} \sigma^{(j)}(\mathcal{F}(x_i; \theta)), \tag{3}$$

The *confidence error* $E_C(s)$ for sample $s$ is defined as the difference between the probability assigned to the predicted label $y_i'$ and the probability associated with the original label $\tilde{y}_i$:

$$E_C(s) = C^{(y_i')} - C^{(\tilde{y}_i)}, \tag{4}$$

where $E_C(s) \in [0, 1]$. In other words, the confidence error states how much the model confidence in the original class is far from the model confidence in the predicted class. The confidence error of zero implies that the original and predicted classes are the same.

## 3.3 Probability of Error

In the following, we theoretically prove the probability that confidence error wrongly identifies noisy labels as clean ones and vice versa is bounded. Presume $H^*$ is a Bayes optimal classifier that predicts the correct label according to the true conditional probability $\mathcal{P}_j(x) = \Pr[y = j|x]$. Consider $v = H^*(x) = \arg \max_j \mathcal{P}_j(x)$ as the $H^*$'s best prediction and $w = \arg \max_{j, j \neq v} \mathcal{P}_j(x)$ as its second best prediction. Define $\tilde{\mathcal{P}}_j(x)$ as the noisy conditional probability, and $\epsilon$ as the maximum approximation error of the classifier $\mathcal{F}$:

$$\tilde{\mathcal{P}}_j(x) = \Pr[\tilde{y} = j|x] = \sum_{l=1}^{k} \Pr[\tilde{y} = j|y = l] * \mathcal{P}_l(x) = \sum_{l=1}^{k} \tau_{lj} * \mathcal{P}_l(x), \quad \epsilon = \max_{x,j} \left[ \left| C^{(j)} - \tilde{\mathcal{P}}_j(x) \right| \right],$$

where $\tau_{lj}$ represents the probability that the label $l$ is flipped to label $j$. Presume the true conditional probability $\mathcal{P}$ meets the multi-class Tsybakov noise condition (Zheng et al., 2020), which guarantees the presence of a margin (region of uncertainty) around the decision boundary separating different classes. This implies that the true conditional probabilities are sufficiently apart, and there is a reasonable level of distinguishability between the classes.

**Lemma 1.** Given the true conditional probability $\mathcal{P}$ satisfying the multi-class Tsybakov noise condition, there exists $\alpha = \min \left\{ 1, \min_x \tau_{\tilde{y}\tilde{y}} \mathcal{P}_w(x) + \sum_{l \neq \tilde{y}} \tau_{l\tilde{y}} * \mathcal{P}_l(x) \right\}$ such that

$$\Pr \left[ \tilde{y} = H^*(x), C^{(\tilde{y})}(x) < \alpha \right] \leq \beta \left[ \mathcal{O}(\epsilon) \right]^\gamma, \tag{5}$$

for constants $\mu \in (0, 1]$, $\beta, \gamma > 0$, and $\epsilon < \mu \min_j \tau_{jj}$.

*Proof.* The proof can be found in Zheng et al. (2020). □

In simple terms, Lemma 1 states that if a label is noisy and the model confidence in that label is low, the label has a limited probability of being correct. The probability of correctness is determined by $\epsilon$, the maximum approximation error for the model, which tends to be small in practical scenarios (Zheng et al., 2020). In other words, Lemma 1 implies that the error bound for model confidence is small in practice.

Now, we provide the error bound for our proposed metric. We consider two possible error cases: (I) the label is noisy according to the optimal Bayes classifier $H^*$, but our metric recognizes it as clean, and (II) the label is clean based on $H^*$, but our metric identifies it as noisy. In the following theorem, we show the probability of making any of these two errors is bounded.

**Theorem 1.** Given that the true conditional probability $\mathcal{P}$ satisfies the multi-class Tsybakov noise condition for constants $\mu \in (0, 1]$, $\beta, \gamma > 0$, and $\epsilon < \mu \min_j \tau_{jj}$, we have:

Case (I): Let the threshold $\alpha = \max_x \left\{ -\sigma^{(\tilde{y})}(x) + \tau_{y'y'} \mathcal{P}_w(x) + \sum_{l, l \neq y'} \tau_{ly'} \mathcal{P}_l(x) \right\}$, then:

$$\Pr \left[ \tilde{y} \neq H^*(x), E_C(x, \tilde{y}) \leq \alpha \right] \leq \beta \left[ O(\epsilon) \right]^\gamma + \psi. \tag{6}$$

Case (II): Let the threshold $\alpha = \min_x \left\{ \sigma^{(y')}(x) - \tau_{\tilde{y}\tilde{y}} \mathcal{P}_w(x) - \sum_{l, l \neq \tilde{y}} \tau_{l\tilde{y}} \mathcal{P}_l(x) \right\}$, then:

$$\Pr \left[ \tilde{y} = H^*(x), E_C(x, \tilde{y}) > \alpha \right] \leq \beta \left[ O(\epsilon) \right]^\gamma. \tag{7}$$

*Proof.* In **Case (I)**, the predicted label is either the same as the Bayes optimal classifier's or not:

$$\begin{aligned} \Pr \left[ \tilde{y} \neq H^*(x), E_C(x, \tilde{y}) \leq \alpha \right] = &\Pr \left[ \tilde{y} \neq H^*(x), E_C(x, \tilde{y}) \leq \alpha, H^*(x) = y' \right] + \\ &\Pr \left[ \tilde{y} \neq H^*(x), E_C(x, \tilde{y}) \leq \alpha, H^*(x) \neq y' \right] \end{aligned} \tag{8}$$

Simplifying the terms and using the definition of $H^*(x)$ yields:

$$\Pr\left[\tilde{y} \neq H^*(x), E_C(x, \tilde{y}) \leq \alpha\right] \leq \Pr\left[\mathcal{P}_w(x) \leq \mathcal{P}_{y'}(x), E_C(x, \tilde{y}) \leq \alpha\right] + \Pr\left[\tilde{y} \neq v, v \neq y'\right] \tag{9}$$

By substituting the definition of confidence error, we have:

$$\Pr\left[\tilde{y} \neq H^*(x), E_C(x, \tilde{y}) \leq \alpha\right] \leq \Pr\left[\mathcal{P}_w(x) \leq \mathcal{P}_{y'}(x), \sigma^{(y')}(x) - \sigma^{(\tilde{y})}(x) \leq \alpha\right] + \Pr\left[\tilde{y} \neq v, v \neq y'\right] \tag{10}$$

Then, we set $\Pr\left[\tilde{y} \neq v, v \neq y'\right] = \psi$ and substitute $\sigma^{(y')}(x)$ with $\tilde{\mathcal{P}}_{y'}(x) - \epsilon$ based on the definition of maximum approximation error:

$$\Pr\left[\tilde{y} \neq H^*(x), E_C(x, \tilde{y}) \leq \alpha\right] \leq \Pr\left[\mathcal{P}_w(x) \leq \mathcal{P}_{y'}(x), \tilde{\mathcal{P}}_{y'}(x) \leq \alpha + \sigma^{(\tilde{y})}(x) + \epsilon\right] + \psi \tag{11}$$

Next, we expand the $\tilde{\mathcal{P}}_{y'}$ term:

$$\begin{aligned} &\Pr\left[\tilde{y} \neq H^*(x), E_C(x, \tilde{y}) \leq \alpha\right] \leq \\ &\Pr\left[\mathcal{P}_w(x) \leq \mathcal{P}_{y'}(x), \left(\tau_{y'y'}\mathcal{P}_{y'}(x) + \sum_{l \neq y'} \tau_{ly'} * \mathcal{P}_l(x)\right) \leq \alpha + \sigma^{(\tilde{y})}(x) + \epsilon\right] + \psi, \end{aligned} \tag{12}$$

and simplify the resulting inequality as follows:

$$\Pr\left[\tilde{y} \neq H^*(x), E_C(x, \tilde{y}) \leq \alpha\right] \leq \Pr\left[\mathcal{P}_w(x) \leq \mathcal{P}_{y'}(x) \leq \frac{\sigma^{(\tilde{y})}(x) + \alpha - \sum_{l \neq y'} \tau_{ly'} * \mathcal{P}_l(x)}{\tau_{y'y'}} + \frac{\epsilon}{\tau_{y'y'}}\right] + \psi \tag{13}$$

Then, we substitute the defined threshold $\alpha$:

$$\Pr\left[\tilde{y} \neq H^*(x), E_C(x, \tilde{y}) \leq \alpha\right] \leq \Pr\left[\mathcal{P}_w(x) \leq \mathcal{P}_{y'}(x) \leq \mathcal{P}_w(x) + \frac{\epsilon}{\tau_{y'y'}}\right] + \psi \tag{14}$$

Utilizing the multi-class Tsybakov noise condition completes the proof for the first case:

$$\Pr\left[\tilde{y} \neq H^*(x), E_C(x, \tilde{y}) \leq \alpha\right] \leq \beta \left[\frac{\epsilon}{\tau_{y'y'}}\right]^{\gamma} + \psi \tag{15}$$

Similarly, we calculate the bound for **Case (II)** as follows:

$$\begin{aligned} \Pr\left[\tilde{y} = H^*(x), E_C(x, \tilde{y}) > \alpha\right] &= \Pr\left[\tilde{y} = H^*(x), E_C(x, \tilde{y}) > \alpha, H^*(x) \neq y'\right] + \\ &\quad \Pr\left[\tilde{y} = H^*(x), E_C(x, \tilde{y}) > \alpha, H^*(x) = y'\right]. \end{aligned} \tag{16}$$

Upon simplifying the terms and substituting the definition of confidence error, we have:

$$\begin{aligned} &\Pr\left[\tilde{y} = H^*(x), E_C(x, \tilde{y}) > \alpha\right] \leq \\ &\Pr\left[\mathcal{P}_w(x) \leq \mathcal{P}_{\tilde{y}}(x), \sigma^{(y')} - \sigma^{(\tilde{y})} > \alpha\right] + \Pr\left[\tilde{y} = H^*(x) = y', \sigma^{(y')} - \sigma^{(\tilde{y})} > \alpha\right] \end{aligned} \tag{17}$$

Substituting the definition of $\epsilon$ yields the following expression:

$$\Pr\left[\tilde{y} = H^*(x), E_C(x, \tilde{y}) > \alpha\right] \leq \Pr\left[\mathcal{P}_w(x) \leq \mathcal{P}_{\tilde{y}}(x), \sigma^{(y')} - \alpha > \sigma^{(\tilde{y})} \geq \tilde{\mathcal{P}}_{\tilde{y}}(x) - \epsilon\right] + 0 \tag{18}$$

Then, we expand the $\tilde{\mathcal{P}}_{\tilde{y}}$ term,

$$\begin{aligned} &\Pr\left[\tilde{y} = H^*(x), E_C(x, \tilde{y}) > \alpha\right] \leq \\ &\Pr\left[\mathcal{P}_w(x) \leq \mathcal{P}_{\tilde{y}}(x), \sigma^{(y')} - \alpha > \sigma^{(\tilde{y})} \geq \tau_{\tilde{y}\tilde{y}}\mathcal{P}_{\tilde{y}}(x) + \sum_{l \neq \tilde{y}} \tau_{l\tilde{y}} * \mathcal{P}_l(x) - \epsilon\right] \end{aligned} \tag{19}$$

and simplify it as follows:

$$\Pr\left[\tilde{y} = H^*(x), E_C(x, \tilde{y}) > \alpha\right] \leq \Pr\left[\mathcal{P}_w(x) \leq \mathcal{P}_{\tilde{y}}(x) \leq \frac{\sigma^{(y')}(x) - \alpha - \sum_{l \neq \tilde{y}} \tau_{l\tilde{y}} * \mathcal{P}_l(x)}{\tau_{\tilde{y}\tilde{y}}} + \frac{\epsilon}{\tau_{\tilde{y}\tilde{y}}}\right] \quad (20)$$

Substituting the threshold $\alpha$ based on its definition and employing the multi-class Tsybakov noise condition results in the following inequality:

$$\Pr\left[\tilde{y} = H^*(x), E_C(x, \tilde{y}) > \alpha\right] \leq \beta \left[\frac{\epsilon}{\tau_{\tilde{y}\tilde{y}}}\right]^\gamma, \quad (21)$$

which completes the proof of Theorem 1. $\qquad\square$

This theorem establishes that the probability of error in distinguishing between clean and noisy samples is bounded if we utilize the confidence error as the sieving criterion. In the following, we present CONFES, which capitalizes on the confidence error metric for sieving the samples in label noise scenarios.

### 3.4 CONFES algorithm

Previous studies (Bai et al., 2021; Liu et al., 2020) show that deep neural networks tend to memorize noisy samples, which can have a detrimental effect on the model utility. Therefore, it is crucial to detect the noisy samples and alleviate their adverse impact, especially in the early steps of training. The CONFES algorithm takes this into consideration by sieving the training samples using the confidence error metric and *completely excluding* the identified noisy samples during training. CONFES (Algorithm 1) consists of three main steps at each epoch: (1) Sieving samples, (2) building the refined training set, and (3) training the model.

---

**Algorithm 1:** Confidence error based sieving (CONFES)

---

**Input:** Noisy training dataset $\tilde{D} = \{(x_i, \tilde{y}_i)\}_{i=1}^n$, model $\mathcal{F}_\theta$, number of training epochs $T$, initial sieving threshold $\alpha$, number of warm-up epochs $T_w$, batch size $m$
**Output:** Trained model $\mathcal{F}_\theta$

1 **for** $i = 0, \ldots, T - 1$ **do**
2  $\quad \alpha_i = \max(\alpha - i \cdot \frac{\alpha}{T_w}, 0)$ /* Set sieving threshold                                                   */
3  $\quad D_i^c = \{s \in \tilde{D} \mid E_C(s) \leq \alpha_i\}$ /* Compute confidence error using equation 4 and sieve clean samples    */
4  $\quad D_i^a = D_i^c \oplus \{(x_j, \tilde{y}_j) \in D_i^c \ s.t. \ j = 1, \ldots, size(\tilde{D}) - size(D_i^c)\}$ /* Build new dataset(clean⊕duplicate)  */
5  $\quad$ **for** $mini\text{-}batch \ \beta = \{(x_j, \tilde{y}_j)\}_{j=1}^m \in D_i^a$ **do**
   $\qquad$ /* Train the model on new dataset                                                                          */
6  $\qquad$ Update model $\mathcal{F}_\theta$ on mini batch $\beta$ using equation 1

7 **return** Trained model $\mathcal{F}_\theta$

---

In the sieving step, the confidence error for each training sample is computed using Equation 4; then, the samples whose confidence error is less than or equal to $\alpha_i$ (sieving threshold at epoch $i$) are considered as clean, whereas the remaining samples are assumed to be noisy and excluded from training. The per-epoch sieving threshold $\alpha_i$ is computed using two hyper-parameters: the initial sieving threshold $\alpha$, and the number of warm-up epochs $T_w$, where $\alpha_i$ is linearly reduced from $\alpha$ to zero during $T_w$ warm-up epochs. The idea of adaptive sieving threshold $\alpha_i$ is based on the observation that generalization occurs in the initial epochs of training, while memorization gradually unfolds afterward (Stephenson et al., 2021; Liu et al., 2020). We capitalize on the warm-up mechanism using an adaptive sieving threshold by training the model on a carefully selected subset of samples, which are potentially clean labels according to their confidence error values, instead of using all samples *from the beginning* of the training, laying a solid foundation for the learning process.

In the second step, a new training dataset is created by concatenating ($\oplus$) only the identified clean samples and their augmentations (duplicates) such that this dataset becomes as large as the initial training set. This is based on the fact that sieving the clean samples results in a reduction in the number of training samples

due to the exclusion of noisy samples. Duplication of the clean samples accounts for this reduction and emphasizes learning the potentially clean samples. Moreover, in line with a previous study (Carlini et al., 2023), which indicates duplication is a very strong promoter of learning, duplicating clean samples produces a very strong learning signal which improves the algorithm overall. Finally, the model is trained on this augmented dataset.

## 4 Evaluation

We draw a performance comparison between CONFES and recent baseline approaches on three label noise settings: symmetric, pairflip, and instance-dependent. In the following, we first describe the experimental setting and then present and discuss the comparison results. Moreover, we provide additional results regarding the effectiveness of the confidence error metric and CONFES algorithm in sieving the samples as well as the sensitivity of CONFESS to its hyper-parameters.

### 4.1 Experimental Setup

**Datasets.** We utilize the CIFAR-10/100 datasets (Krizhevsky et al., 2009) and make them noisy using different types of synthetic label noise. Furthermore, we incorporate the Clothing1M dataset (Xiao et al., 2015), a naturally noisy benchmark dataset widely employed in previous studies. CIFAR-10/100 contain 50000 training samples and 10000 testing samples of shape $32 \times 32$ from 10/100 classes. For the CIFAR datasets, we perturb the training labels using symmetric, pairflip, and instance-dependent label noise introduced in Xia et al. (2020), but keep the test set clean. Following data augmentation/preprocessing procedure in previous works (Liu et al., 2020; Li et al., 2020), the training samples are horizontally flipped with probability 0.5, randomly cropped with size $32 \times 32$ and padding $4 \times 4$, and normalized using the mean and standard deviation of the dataset. Clothing1M is a real-world dataset of 1 million images of size $224 \times 224$ with noisy labels (whose estimated noise level is approximately 38% (Wei et al., 2022; Song et al., 2019)) and 10k clean test images from 14 classes. Following prior studies (Liu et al., 2020; Li et al., 2020), the data augmentation steps performed on the clothing1M dataset include $256 \times 256$ resizing, $224 \times 224$ random cropping, and random horizontal flipping. In clothing1M, the number of samples for each class is imbalanced. We follow Li et al. (2020) and sample a class-balanced subset of the training dataset at each epoch.

**State-of-the-art methods.** On all considered datasets, we compare CONFES with the most recent related studies including (1) standard cross-entropy loss (CE), (2) Co-teaching (Han et al., 2018) that cross-trains two models and uses the small-loss trick for selecting clean samples and exchanges them between the two models, (3) ELR (Liu et al., 2020), an early-learning regularization method that leverages the model output during the early-learning phase, (4) CORES$^2$ (Cheng et al., 2020), a sample sieving approach that uses confidence regularization which leads the model towards having more confident predictions, (5) PES (Bai et al., 2021), a progressive early-stopping strategy, (6) SLN (Chen et al., 2021a) that improves regularization by introducing stochastic label noise, and (7) LRT, a confidence based algorithm that leverages likelihood ratio values for sample selection. Co-teaching, CORES$^2$, and LRT are based on sample sieving, whereas ELR and SLN are regularization-based methods. For all methods, the specific hyper-parameters are set according to the corresponding manuscript or the published source code if available.

**Parameter Settings and Computational Resources.** We conduct the experiments on a single GPU system equipped with an NVIDIA RTX A6000 graphic processor and 48GB of GPU memory. Our method is implemented in PyTorch v1.9. For all methods, we evaluate the average test accuracy on the last five epochs, and for co-teaching, we report the average of this metric for the two networks. Following previous works (Li et al., 2020; Bai et al., 2021), we train the PreActResNet-18 (He et al., 2016) model on CIFAR-10 and CIFAR-100 using the SGD optimizer with momentum of 0.9, weight decay of 5e-4, and batch size of 128. The initial learning rate is set to 0.02, which is decreased by 0.01 in 300 epochs using cosine annealing scheduler (Loshchilov & Hutter, 2017). For the Cloting1M dataset, we adopt the setting from Li et al. (2020) and train the ResNet-50 model for 80 epochs. The optimizer is SGD with momentum of 0.9 and weight decay of 1e-3. The initial learning rate is 0.002, which is reduced by factor of 10 at epoch 40. At each

epoch, the model is trained on 1000 mini-batches of size 32. Note that ResNet-50 has been pretrained on ImageNet (Deng et al., 2009).

### 4.2 Results

**CIFAR-10/100 datasets.**   Tables 2 and 3 list test accuracy values for different noise types and noise rates on CIFAR-10 and CIFAR-100 datasets respectively. According to these tables, CONFES outperforms the competitors for all considered symmetric, pairflip, and instance-dependent noise types. Similarly, CONFES delivers higher accuracy than the competitors for different noise rates. Moreover, as the noise level increases, the accuracy gap between CONFES and its competitors widens in favor of CONFES. Figure 1 illustrates the test accuracy versus epoch for the different learning algorithms. As shown in the figure, CONFES is robust against overfitting because the corresponding test accuracy continues to increase as training moves forward, and stays at the maximum after the model converges. Some of the other algorithms such as SLN and ELR, on the other hand, suffer from the overfitting problem, where their final accuracy values are lower than the maximum accuracy they achieve.

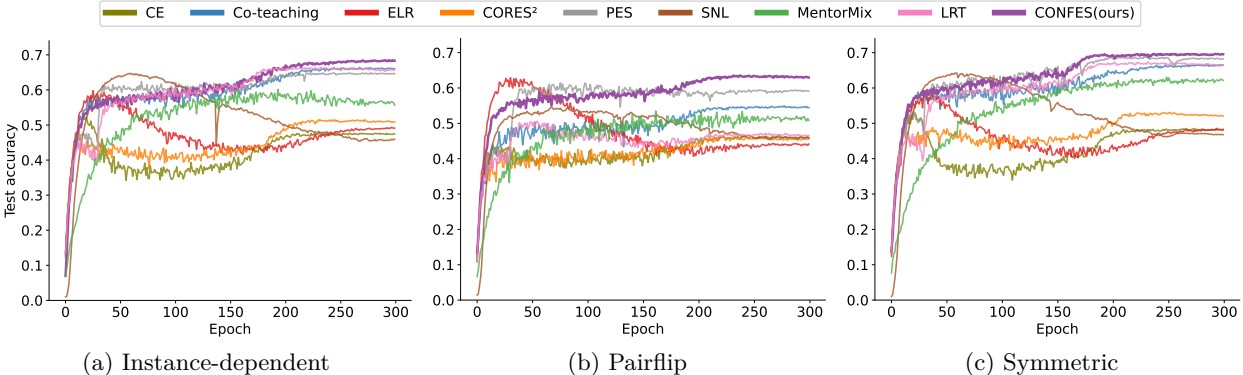

Figure 1: **Test accuracy** for **PreAct-ResNet18** trained on **CIFAR-100**: CONFES is robust against overfitting, whereas some competitors including SLN and ELR suffer from overfitting; noise level is 40%.

Table 2: Test accuracy on **CIFAR-10** for different noise types with noise level 40%.

| Method | Symmetric | Pairflip | Instance |
|---|---|---|---|
| CONFES (ours) | **90.62**±0.2 | **86.18**±0.3 | **90.28**±0.2 |
| CE | 66.61 ±0.4 | 59.25 ±0.1 | 66.04 ±0.2 |
| Co-teaching (Han et al., 2018) | 87.42 ±0.2 | 84.57 ±0.2 | 86.90±0.1 |
| ELR (Liu et al., 2020) | 85.74 ±0.2 | 86.15 ±0.1 | 85.37 ±0.3 |
| CORES$^2$ (Cheng et al., 2020) | 83.90 ±0.4 | 58.38 ±0.6 | 76.71 ±0.4 |
| LRT (Zheng et al., 2020) | 85.47 ±0.3 | 59.25 ±0.3 | 80.53 ±0.9 |
| PTD (Xia et al., 2020) | 72.05 ±0.9 | 58.34 ±0.8 | 65.97 ±0.9 |
| PES (Bai et al., 2021) | 90.55 ±0.1 | 85.56 ±0.1 | 85.63 ±0.5 |
| SLN (Chen et al., 2021a) | 83.69 ±0.2 | 85.26 ±0.5 | 67.71 ±0.4 |

**Combining CONFES with state-of-the-art algorithms.**   Table 4 shows the accuracy values of CoTeaching, JoCor, and DivideMix if confidence error is used as the discriminator metric instead of the training loss. As shown in the table, the accuracy from these algorithms is enhanced by 2-5% compared to their baseline performance by combining them with CONFES, indicating that confidence error is not only effective as the main building block of the proposed CONFES algorithm but also combined with other state-of-the-art methods including DivideMix, which is a complex method employing data augmentation, and guessing or refining the noisy labels rather than excluding them, which helps in utilizing the noisy samples and learning their feature information.

Table 3: Test accuracy on **CIFAR-100** for various label noise types with different noise rates.

(a) **Instance-dependent**

| Method | 20% | 40% | 60% |
|---|---|---|---|
| CONFES (ours) | **73.59**±0.2 | **69.68**±0.2 | **59.48**±0.1 |
| CE | 63.16 ±0.1 | 48.92 ±0.3 | 30.65 ±0.4 |
| Co-teaching (Han et al., 2018) | 71.12 ±0.3 | 66.55 ±0.3 | 57.18 ±0.2 |
| ELR (Liu et al., 2020) | 63.10 ±0.2 | 49.15 ±0.2 | 29.88 ±0.6 |
| CORES$^2$ (Cheng et al., 2020) | 64.55 ±0.1 | 50.98 ±0.2 | 33.93 ±0.5 |
| LRT (Zheng et al., 2020) | 73.14 ±0.2 | 65.32 ±0.6 | 45.37 ±0.1 |
| MentorMix (Jiang et al., 2020) | 69.41 ±0.2 | 56.41 ±0.1 | 34.61 ±0.1 |
| PES (Bai et al., 2021) | 71.65 ±0.3 | 64.83 ±0.2 | 41.10 ±0.5 |
| SLN (Chen et al., 2021a) | 60.08 ±0.1 | 46.08 ±0.3 | 29.77 ±0.4 |

(b) **Pairflip**

| Method | 20% | 30% | 40% |
|---|---|---|---|
| CONFES (ours) | **73.12**±0.1 | **71.34**±0.2 | **62.37**±0.4 |
| CE | 64.31 ±0.3 | 55.77±0.1 | 45.62 ±0.4 |
| Co-teaching (Han et al., 2018) | 69.59 ±0.2 | 64.04 ±0.4 | 55.42 ±0.5 |
| ELR (Liu et al., 2020) | 62.05 ±0.5 | 54.44 ±0.2 | 44.31 ±0.3 |
| CORES$^2$ (Cheng et al., 2020) | 63.85 ±0.2 | 54.88 ±0.3 | 45.34±0.2 |
| LRT (Zheng et al., 2020) | 71.70 ±0.1 | 60.78 ±0.1 | 46.24 ±0.2 |
| MentorMix (Jiang et al., 2020) | 69.65 ±0.1 | 62.01 ±0.1 | 50.97 ±0.2 |
| PES (Bai et al., 2021) | 71.73 ±0.4 | 68.28 ±0.3 | 59.18 ±0.2 |
| SLN (Chen et al., 2021a) | 61.82 ±0.3 | 53.67 ±0.2 | 45.72 ±0.2 |

(c) **Symmetric**

| Method | 20% | 40% | 60% |
|---|---|---|---|
| CONFES (ours) | **73.89**±0.1 | **69.63**±0.2 | **60.65**±0.1 |
| CE | 63.46 ±0.7 | 47.85 ±0.4 | 29.59 ±0.3 |
| Co-teaching (Han et al., 2018) | 71.54 ±0.3 | 66.26 ±0.1 | 58.82 ±0.1 |
| ELR (Liu et al., 2020) | 63.59 ±0.1 | 48.33 ±0.2 | 30.37 ±0.1 |
| CORES$^2$ (Cheng et al., 2020) | 65.99 ±0.5 | 52.26 ±0.2 | 34.61 ±0.2 |
| LRT (Zheng et al., 2020) | 73.72 ±0.1 | 66.52 ±0.2 | 50.86 ±0.4 |
| MentorMix (Jiang et al., 2020) | 71.52 ±0.2 | 61.96 ±0.2 | 44.38 ±0.3 |
| PES (Bai et al., 2021) | 71.42 ±0.2 | 68.37 ±0.2 | 60.38 ±0.1 |
| SLN (Chen et al., 2021a) | 60.48 ±0.1 | 46.98 ±0.2 | 28.50 ±0.2 |

Table 4: Test accuracy for CONFES combined with other approaches on CIFAR-100 with noise rate 40%.

| Method | Symmetric | Pairflip | Instance |
|---|---|---|---|
| Co-teaching (Han et al., 2018) | 66.26 ±0.1 | 55.42 ±0.5 | 66.55±0.3 |
| CONFES-Co-teaching | 69.94±0.1 | 57.90±0.2 | 69.51±0.1 |
| **Improvement** | **+3.68** | **+2.48** | **+2.96** |
| DivideMix (Li et al., 2020) | 74.63±0.2 | 74.9±0.1 | 66.79±0.3 |
| CONFES-DivideMix | 76.31±0.2 | 76.51±0.1 | 69.03±0.1 |
| **Improvement** | **+1.68** | **+1.61** | **+2.24** |
| JoCoR (Wei et al., 2020) | 67.05±0.2 | 54.96±0.3 | 67.46±0.2 |
| CONFES-JoCoR | 70.48±0.2 | 59.61±0.1 | 70.24±0.4 |
| **Improvement** | **+3.43** | **+4.65** | **+2.78** |

**Clothing1M dataset.** Table 5 summarises the performance of methods on Clothing1M dataset. CORES$^2$ and PES provide slight or no accuracy gain compared to the baseline cross-entropy training. CONFES, on the other hand, outperforms the competitors including ELR and SLN.

Table 5: Test accuracy on **Clothing1M** dataset

| Method | CE | ELR | CORES$^2$ | PES | SLN | CONFES (ours) |
|---|---|---|---|---|---|---|
| Test accuracy | 69.21% | 71.39% | 69.50% | 69.18% | 72.80% | **73.24%** |

**Effectiveness of confidence error.** We design an experiment to illustrate the effectiveness of the confidence error metric: we employ the SGD optimizer and cross-entropy loss function to train PreActResNet18 on CIFAR-100, where 40% of the labels are made noisy using the instance-dependent noise. At the beginning of each epoch, the model computes the confidence error for all training samples and sorts them in ascending order by the confidence error value. The model considers the first 60% of the samples with lower confidence error values as clean and only incorporates them during training. This procedure is repeated for 200 epochs. As shown in Figure 2, the distribution of confidence error values for the clean and noisy samples becomes more and more dissimilar as the training process proceeds. For instance, at epoch 50, a sample with a high confidence error (e.g. near 1.0) is much more likely to be a noisy sample than a clean one. Likewise, a sample with a very low confidence error is probably clean. The extensions of this experiment to pairflip and symmetric label noise are available at Figures 6-7 in the Appendix. These observations are highly consistent with the theoretical analysis from Theorem 1, which states the probability of error (identifying noisy labels wrongly as clean and vice versa) for the confidence error metric is bounded and low in practice.

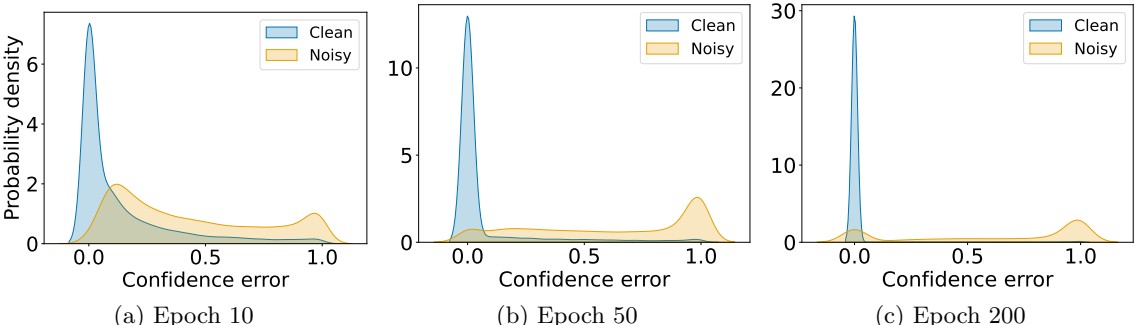

(a) Epoch 10        (b) Epoch 50        (c) Epoch 200

Figure 2: **Distributions of confidence error values** for clean and noisy samples progressively diverge from each other as the training process continues. The experiment is conducted using PreAct-ResNet18 and CIFAR-100 with noise level of 40%.

**Why CONFES?** We use our previous experimental setup and train the model with the naive cross-entropy method and CONFES algorithm to answer this question. Figures 3a and 3b show the model confidence for the noisy, clean, and predicted labels (averaged over the corresponding samples) with cross-entropy and CONFES, respectively. According to Figure 3a, the confidence over noisy labels is very low at the early stages of cross-entropy training. However, as the training proceeds, the model's confidence in noisy labels increases. At the end of the training, the model confidence over predicted and noisy labels is close to each other. This indicates that the model has been misled by the noisy samples, wrongly considering them as the true labels of the samples. CONFES, on the other hand, utilizes the model confidence error to distinguish between clean and noisy samples and exclude the identified noisy samples during training. This results in consistently low confidence for the noisy samples, but high confidence in clean and predicted labels throughout all training stages, as shown in Figure 3b. This observation shows the importance of identifying noisy samples efficiently and keeping confidence in them as low as possible as performed by the CONFES algorithm.

Furthermore, we employ the CONFES algorithm in the same setting as Figures 2 and 3 to calculate the confusion matrix and empirically examine the error made by CONFES in differentiating between the clean and noisy labels in practice. Figure 4 shows the confusion matrix for the CONFES algorithm. According to the figure, CONFES is effective in recognizing the noisy samples from the beginning to the end of the training, where it correctly identifies around 38% out of 40% of noisy samples. On the other hand, the algorithm wrongly identifies many clean samples as noisy in the early epochs (around 27%). However, as training moves forward, CONFES becomes more and more efficient in identifying the clean samples, where it correctly recognizes around 55% out of 60% of the clean samples at the last epoch. Figure 8 in the Appendix, moreover, visualizes the number of clean and noisy samples that CONFES identifies correctly for different noise types and noise rates, which are consistent with those from confusion matrices.

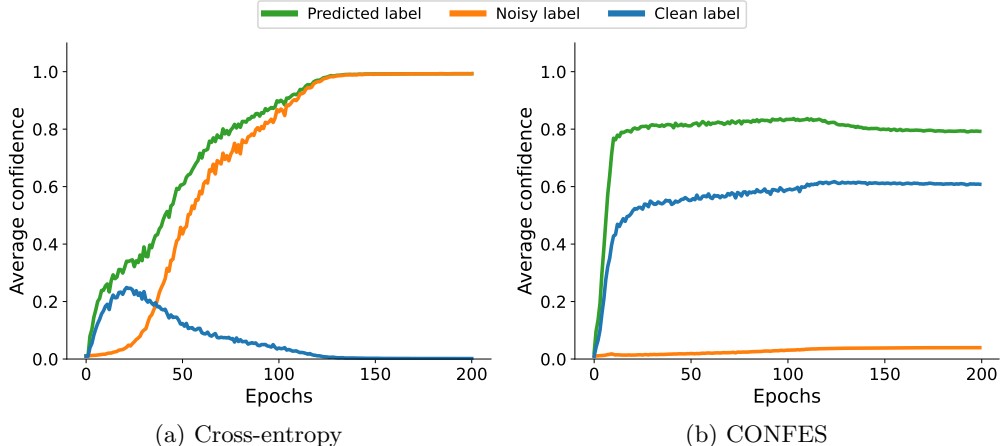

(a) Cross-entropy

(b) CONFES

Figure 3: **Effectiveness of CONFES**: Using naive cross-entropy training (a), the model confidence over noisy labels increases as the training moves forward. It implies that the model is misled by the noisy samples. CONFES (b), however, differentiates the noisy samples from the clean ones and excludes the identified noisy samples during training. This leads to very low model confidence over the noisy samples but high confidence in clean and predicted labels throughout training.

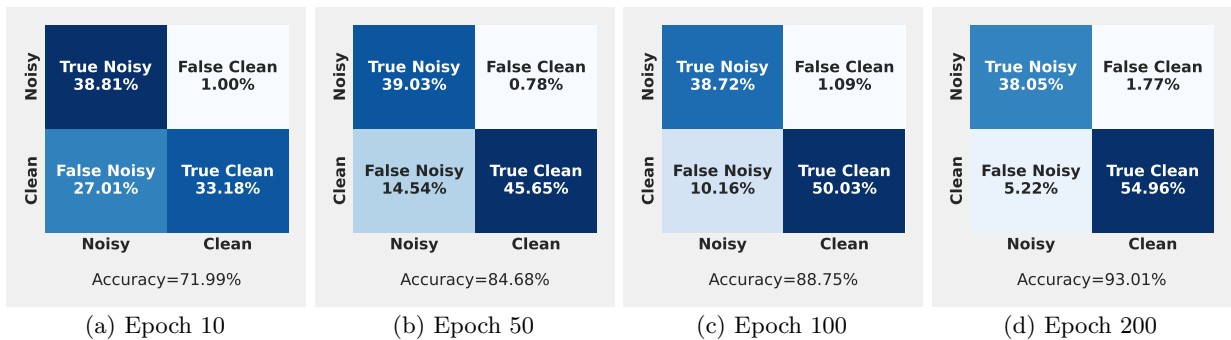

(a) Epoch 10        (b) Epoch 50        (c) Epoch 100        (d) Epoch 200

Figure 4: **Confusion matrix for the CONFES algorithm**: In early epochs, CONFES correctly identifies the majority of noisy labels (around 38% out of 40%), but wrongly identifies many clean labels as noisy ones (about 27%). As training proceeds, the algorithm not only still remains effective in identifying the noisy labels (around 38% out of 40%) but also correctly recognizes the clean labels (about 55% out of 60%). The model is PreActResNet18 trained on CIFAR-100 with instance-dependant label noise of rate 40%.

**Sensitivity analysis of hyper-parameters.** The initial sieving threshold $\alpha$ and number of warm-up epochs $T_w$ are the hyper-parameter of the proposed CONFES algorithm. The per-epoch sieving threshold is computed using the aforementioned hyper-parameters. For CIFAR-100, we set $\alpha$=0.2 and $T_w$=30 for all noise types and noise rates. For CIFAR-10, the values of $\alpha$ and $T_w$ are 0.1 and 25, respectively, for symmetric and instance-dependent noise types. For Clothing1M, $\alpha$ and $T_w$ are set to 0.05 and 3, respectively. We also investigate the sensitivity of CONFES to its hyper-parameters using the CIFAR-100 dataset with noise rate of 40% for symmetric, instance-dependent, and pairflip noise settings. To analyze the sensitivity to $T_w$, we set $\alpha = 0.2$ and use four different values for warm-up epochs: $T_w \in \{5, 20, 30, 50\}$. Similarly, we set $T_w = 30$ and employ four different values for sieving threshold: $\alpha \in \{0.1, 0.2, 0.3, 0.5\}$. As shown in Figure 5, the accuracy reductions using the suboptimal hyper-parameter values compared to the optimal setting ($\alpha = 0.2$ and $T_w = 30$) are 1.6%, 2.3% and 4.1% for symmetric, instance-dependent, and pairflip noise settings, respectively, at the worst case. This indicates that CONFES is relatively robust against hyper-parameter value choices, making it easy to employ or tune by practitioners.

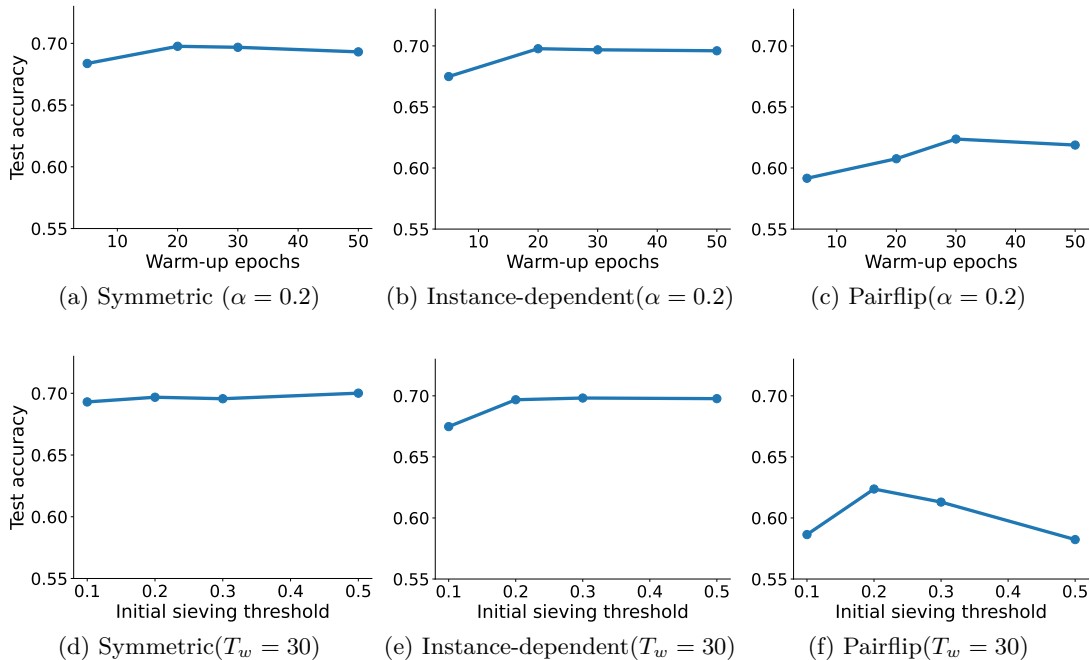

(a) Symmetric ($\alpha = 0.2$)     (b) Instance-dependent($\alpha = 0.2$)     (c) Pairflip($\alpha = 0.2$)

(d) Symmetric($T_w = 30$)     (e) Instance-dependent($T_w = 30$)     (f) Pairflip($T_w = 30$)

Figure 5: **Sensitivity analysis of CONFES to its hyper-parameters** $T_w$(number of warm-up epochs) and $\alpha$ (initial sieving threshold) for different noise types. The dataset is CIFAR-100 with a noise rate of 40%.

## 5    Discussion and Conclusion

We present the confidence error metric to effectively discriminate between noisy and clean samples in label noise learning settings. Moreover, we theoretically prove the probability of error for the proposed metric is bounded and experimentally show it is small in practice. We integrate the confidence error metric into a learning algorithm called CONFES, which refines the training samples by keeping only the identified clean samples and filtering out the noisy ones. Our experimental results verify the robustness of CONFES under different noise types such as symmetric, pairflip, and instance-dependent, especially when noise levels are high. We also demonstrate that confidence error can be employed by other algorithms including Co-teaching and DivideMix to further improve the model accuracy.

**CONFES versus baseline methods.** According to the experimental results, CONFES outperforms all baseline methods in the considered symmetric, pairflip, and instant-dependent noise settings. As the noise rate increases, the efficiency of the CONFES algorithm becomes more apparent (e.g., noise rate of 60% in CIFAR-100). Moreover, CONFES is robust to overfitting unlike some of its competitors such as SLN and ELR. In terms of computational overhead, CONFES has one additional forward pass for constructing the refined dataset, which only includes clean samples according to the confidence error metric. However, methods such as Co-teaching (Han et al., 2018) employ two networks in the training process, which makes them substantially less computationally efficient compared to our approach.

Although some methods such as PES (Bai et al., 2021) perform well in the presence of symmetric label noise, their accuracy decreases in more complex noise settings such as instance-dependent, which is not the case for CONFES. Additionally, the accuracy of some other baseline methods such as LRT (Zheng et al., 2020) and MentorMix (Jiang et al., 2020) drastically reduces in a highly noisy setting (e.g., with 60% noise rate). Approaches such as ELR (Liu et al., 2020) and PES (Bai et al., 2021) work well for "easy to classify" datasets such as CIFAR-10, but their efficiency reduces on more challenging datasets including CIFAR-100 and Clothing1M. CONFES, on the other hand, outperforms the compared baselines in different noise types (symmetric, instance-dependent, and pairflip), with various noise levels (i.e., 20%, 40% and 60%), and on CIFAR-10, CIFAR-100 and the challenging Clothing1M datasets.

**CONFES versus LRT.** Our work is related to the work from Zheng et al. (2020) which proposed a confidence-based metric called likelihood ratio test (LRT) for sieving the clean samples. Our proposed confidence error metric has at least two advantages over the likelihood ratio: (1) Confidence error enables the algorithm to start performing the sample sieving in the early epochs of training. Using the sieving threshold $\alpha_i$, the algorithm only incorporates the samples with confidence error less than $\alpha_i$ in the training instead of all samples. Applying a similar threshold to likelihood ratio in warm-up epochs delivers much lower accuracy than using all samples based on our observations. (2) The confidence error is a more efficient metric than the likelihood ratio for differentiating the clean samples from the noisy ones according to our experimental results provided in Figure 10 in the Appendix, which are indeed consistent with the accuracy results provided in the Evaluation section. Moreover, we empirically compared the probability of error for confidence error and LRT on the CIFAR-100 dataset with different types and levels of noise. The results (Figure 9 in the Appendix) show confidence error has a much smaller error rate in identifying noisy samples compared to LRT, while its error rate in identifying clean samples is slightly worse than LRT. In sample sieving, misidentifying noisy samples as clean ones (false negatives) is much more detrimental to utility than wrongly recognizing clean samples as noisy (false positives). The latter issue can be alleviated with techniques such as clean data duplication as employed by CONFES.

In the future, we can extend our work by incorporating techniques such as semi-supervised learning to perform label correction. We can also automate the selection process for sample sieve size by utilizing soft clustering techniques to model confidence errors. This approach would eliminate the need for the initial sieving threshold hyper-parameter ($\alpha$). Furthermore, we can employ ensemble learning techniques including Adaboost-like methodology. Leveraging the proposed confidence error metric and incorporating multiple weak classifiers might improve the efficiency of sieving but can be computationally expensive.

## Acknowledgement

This work was supported by a Google Ph.D. Fellowship to R.T., as well as the German Federal Ministry of Education and Research and the Bavarian State Ministry for Science and the Arts. The authors of this work take full responsibility for its content.

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

# A   Appendix

**Further experiments on the effectiveness of confidence error.** We extended the experiments associated with Figure 2 of the main manuscript to the symmetric and pairflip label noise. Experiments are conducted using PreAct-ResNet18 and CIFAR-100 with noise level of 40%.

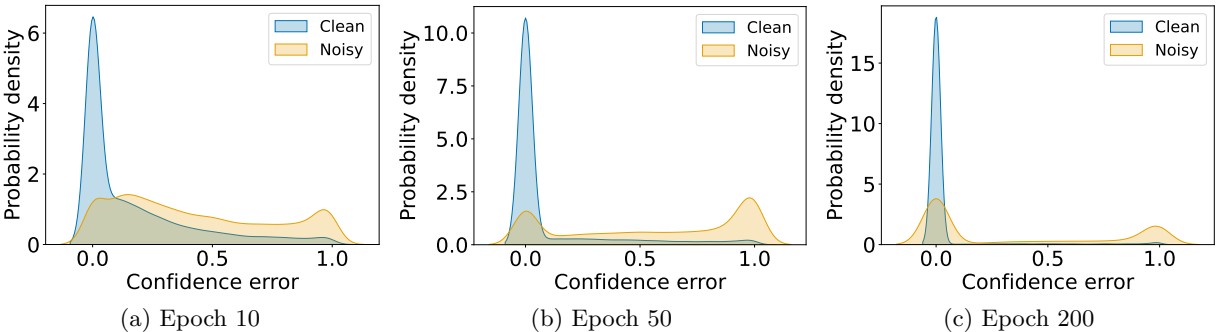

Figure 6: Distributions of **confidence error** values for **pairflip** label noise

.

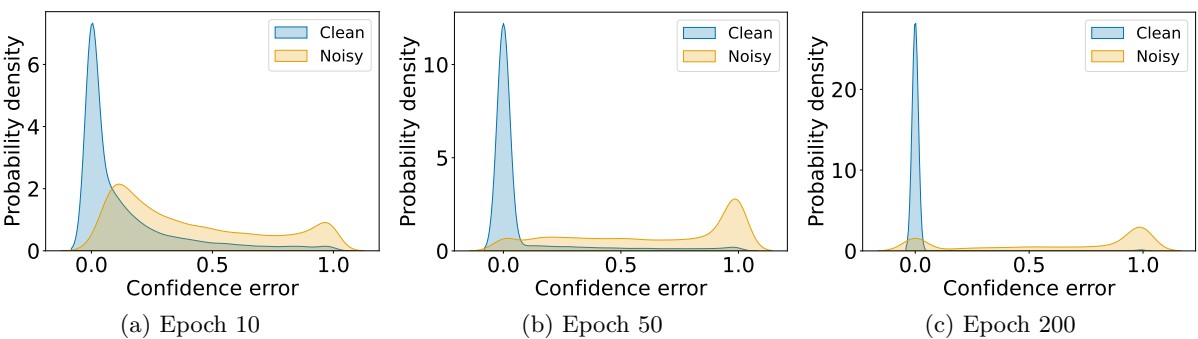

Figure 7: Distributions of **confidence error** values for **symmetric** label noise

**Additional details on the experimental setup.** For all experiments on the CIFAR-10, CIFAR-100, and Clothing1M datasets, there are some general hyper-parameters such as learning rate, batch size, and weight decay, specified in the original manuscript and are summarized in Table 6. The method-specific hyper-parameters used in the experiments are set based on the corresponding manuscript or the published source code: Co-teaching (Han et al., 2018)[*], ELR (Liu et al., 2020) [†], CORES$^2$ (Cheng et al., 2020)[‡], PES (Bai et al., 2021)[§], SLN (Chen et al., 2021a) [¶], DivideMix (Li et al., 2020)[‖], JoCoR (Wei et al., 2020) [**], LRT (Zheng et al., 2020)[††], MentorMix (Jiang et al., 2020)[‡‡] and PTD (Xia et al., 2020)[§§].

**Instance-dependent label noise.** In order to generate the instance-dependent label noise in the experiments, we followed the previous works Cheng et al. (2020); Yao et al. (2020); Bai et al. (2021); Chen et al. (2021a) and employed the following algorithm proposed in Xia et al. (2020):

---

[*]`https://github.com/bhanML/Co-teaching`

[†]`https://github.com/shengliu66/ELR`

[‡]`https://github.com/UCSC-REAL/cores`

[§]`https://github.com/tmllab/PES`

[¶]`https://github.com/chenpf1025/SLN`

[‖]`https://github.com/LiJunnan1992/DivideMix`

[**]`https://github.com/hongxin001/JoCoR`

[††]`https://github.com/pingqingsheng/LRT`

[‡‡]`https://github.com/LJY-HY/MentorMix_pytorch`

[§§]`https://github.com/xiaoboxia/Part-dependent-label-noise`

---

**Algorithm 2:** Instance-dependent Label Noise Generation taken from Xia et al. (2020)

---

**Input:** Clean samples $\{(x_i, y_i)\}_{i=1}^n$, Noise rate $\tau$
**Output:** Noisy samples $\{(x_i, \tilde{y}_i)\}_{i=1}^n$

**1** Sample instance flip rates $q \in \mathbb{R}^N$ from the truncated normal distribution $\mathcal{N}(\tau, 0.1^2, [0, 1])$
**2** Independently samples $w_1, ..., w_c$ from the standard normal distribution $\mathcal{N}(0, 1^2)$
**3** **for** $i = 0, ..., n$ **do**
**4**   $p = x_i \cdot w_{y_i}$ /* Generate instance dependent flip rate                                          */
**5**   $p_{y_i} = -\infty$ /* control the diagonal entry of the instance-dependent transition matrix            */
**6**   $p = q_i \cdot \text{softmax}(\text{p})$ /* make the sum of the off-diagonal entries of the $y_i$-th row to be $q_i$   */
**7**   $p_{y_i} = 1 - q_i$ /* set the diagonal entry to be $1 - q_i$                                             */
**8** Randomly choose a label from the label space according to the possibilities p as noisy label $y_i$
**9** **return** Noisy samples $\{(x_i, \tilde{y}_i)\}_{i=1}^n$

---

Table 6: General training hyperparameters (common for all methods of comparison).

|  | CIFAR-10 | CIFAR-100 | Clothing1M |
|---|---|---|---|
| Model | PreActResNet-18 | PreActResNet-18 | Pretrained ResNet-50 |
| Batch size | 128 | 128 | 32 |
| Learning rate (lr) | 2e-2 | 2e-2 | 2e-3 |
| lr scheduler | Cosine annealing | Cosine annealing | MultiStep |
| lr decay factor | 100 | 100 | 10 at epoch 40 |
| Weight decay | 5e-4 | 5e-4 | 1e-3 |
| Epochs | 300 | 300 | 80 |

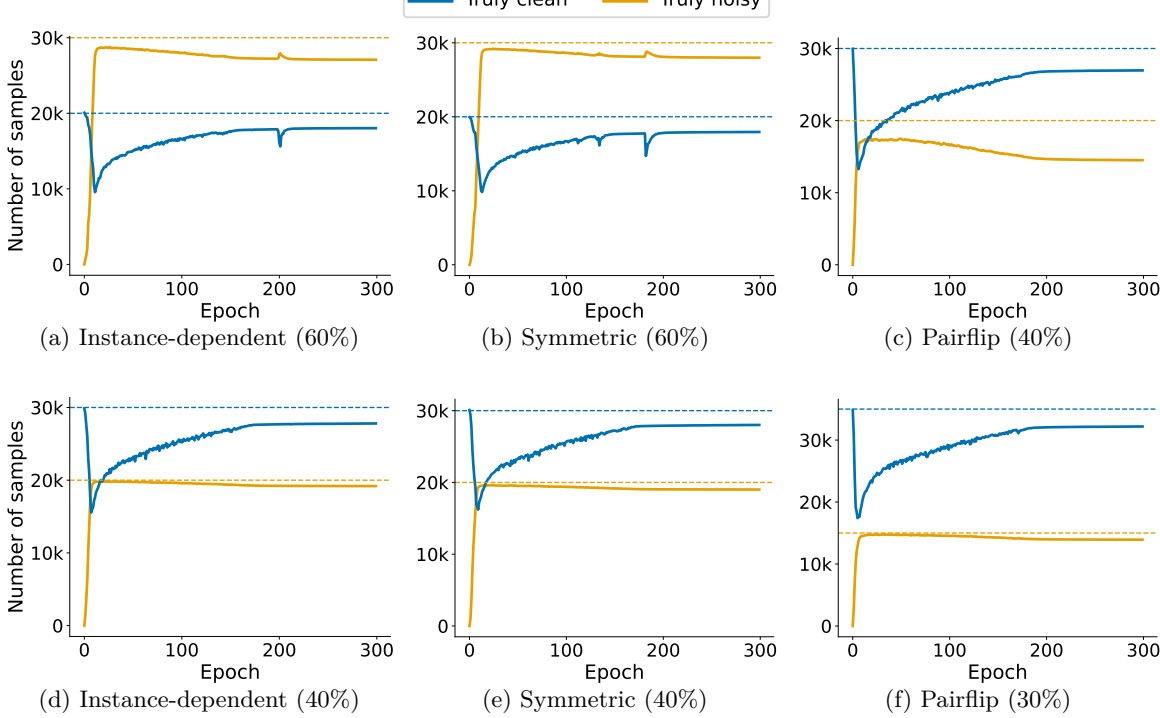

Figure 8: The number of clean and noisy samples that CONFES correctly identifies on CIFAR-100 with different noise types and noise rates. The dashed lines represent the total number of clean and noisy samples. CONFES consistently achieves a high success rate in correctly distinguishing between clean and noisy samples.

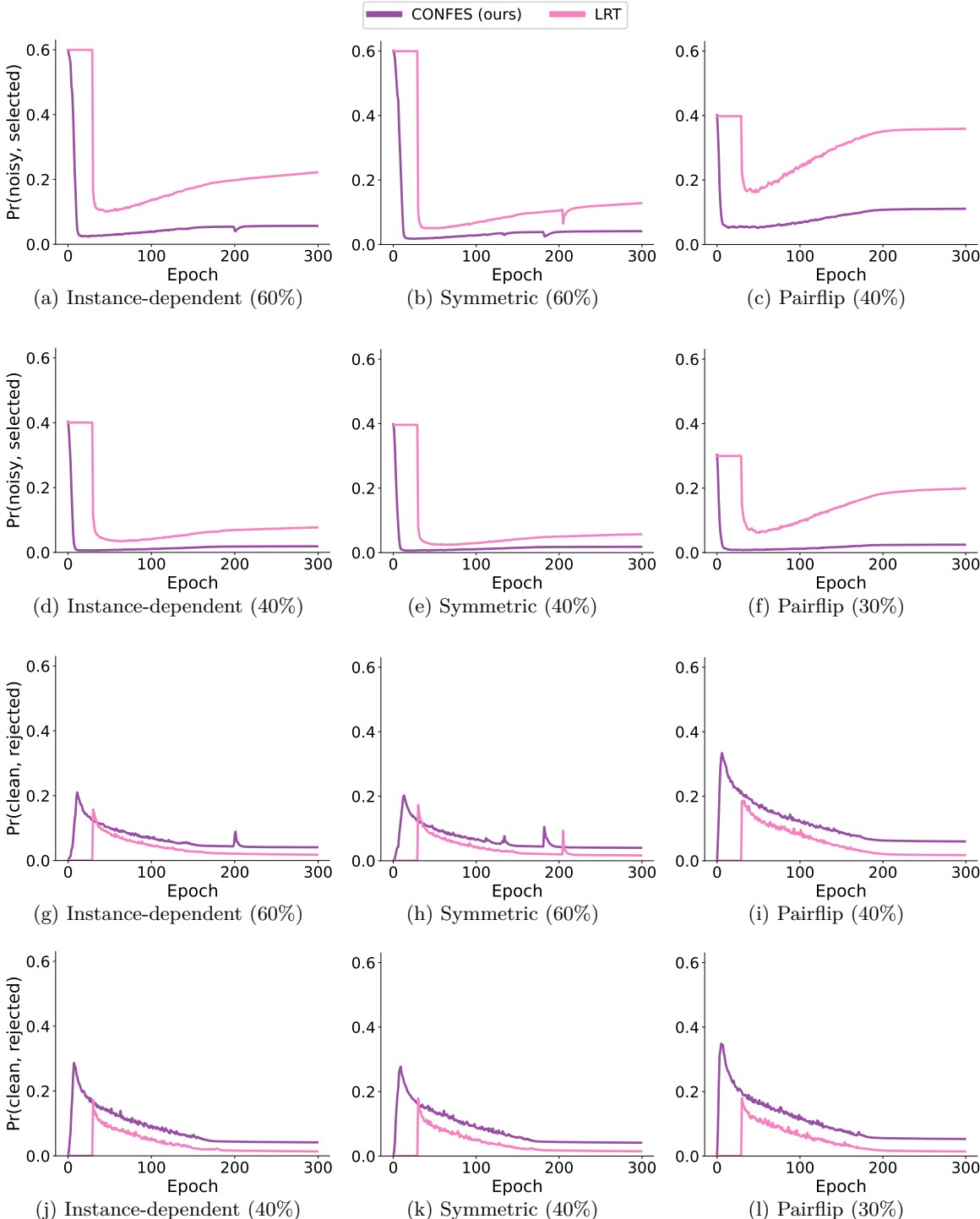

Figure 9: Comparison between the probability of error (lower is better) for CONFES (based on the proposed confidence error metric) and the LRT algorithm (Zheng et al., 2020) (based on the likelihood ratio metric) on the CIFAR-100 dataset with different noise types and noise rates. (noisy, selected) means the sample is noise but wrongly selected by the algorithm to be incorporated in training. Likewise, (clean, rejected) means the sample is clean but excluded by the algorithm during training. The probability of error is lower for CONFES in the former case, whereas it is slightly higher in the latter case. Note that the former case is more detrimental to the performance compared to the latter one.

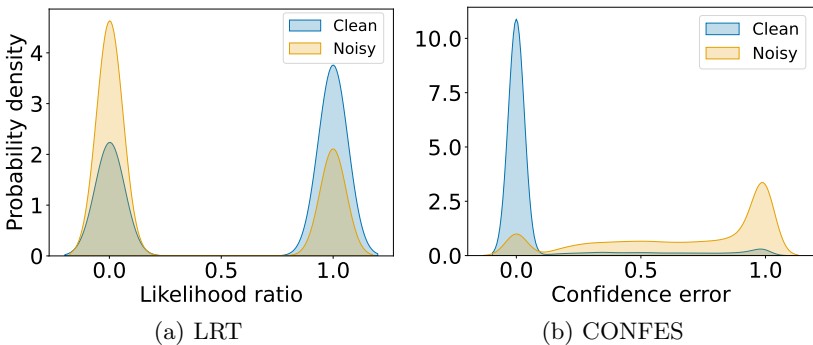

(a) LRT

(b) CONFES

Figure 10: Distributions of **likelihood ratio** employed in LRT(Zheng et al., 2020) and the proposed **confidence error** metric used in CONFES at epoch 200. The distributions of confidence error for noisy and clean samples are more dissimilar than that of likelihood ratio, indicating that confidence error is a more effective metric than likelihood ratio for sieving the samples. The experimental setup is the same as Figure 2.

