# OpenReview forum: "Label Noise-Robust Learning using a Confidence-Based Sieving Strategy"
_TMLR — Accepted by TMLR_

### Review · Reviewer_dBHc · 2023-06-04

**Summary Of Contributions:**

This work leverages an insightful observation to introduce a groundbreaking discriminator metric known as "confidence error" and a filtering technique called CONFES, which significantly enhances the distinction between clean and noisy samples. It also establishes rigorous theoretical assurances regarding the probability of error associated with the novel metric. Furthermore, it conducts comprehensive experimental evaluations, wherein it demonstrates the remarkable performance superiority of its proposed methodology over recent investigations across different scenarios, encompassing both synthetic and real-world label noise.

**Audience:**

Yes

**Broader Impact Concerns:**

None.

**Claims And Evidence:**

Yes

**Requested Changes:**

1. Visualize the number of clean samples the proposed method sieves.
2. Give more explanation to warm-up.
3. Perhaps it would be beneficial to introduce an additional Adaboost-like algorithm to further enhance the novelty of the approach.


**Strengths And Weaknesses:**

Strengths:
1.The paper presents an innovative alternative metric, distinct from previously used loss values, that offers a more effective means of sieving training examples.
2.Theorectically, It also establishes rigorous theoretical assurances regarding the probability of error. Experimentally, the proposed method has an impressive performance and can be equipped into other algorithms.
3.The organization and presentation of the paper are well done.

Weaknesses:
1. It might be better to visualize the number of clean samples the proposed method sieves, since the core of the work is to use a new metric to select clean samples.
2. The reason why the warm-up is used needs to be further explained. Why not only warm up the model and then set the threshold?
3. The contributions seems a little incremental though it compare the method to LRT.

---

### Review · Reviewer_zMtf · 2023-06-28

**Summary Of Contributions:**

The paper studies the problem of noise label learning. The paper proposes a simple metric called the confidence error, which is essentially the difference between the predicted class probability and the label class probability, and uses such a metric to identify clean labels to construct the training data set. The paper shows theoretical properties of the proposed metric that under certain conditions, samples selected using the criteria have bounded probabilities to being noisy. The paper conducts empirical experiments and show that the proposed method achieve better performance compared to some baseline methods.

**Audience:**

Yes

**Claims And Evidence:**

Yes

**Requested Changes:**

Please address my questions and concerns in the questions and weaknesses part.

**Strengths And Weaknesses:**

Strengths:
1. The proposed method is simple and easy to implement.
2. The paper studies theoretical properties of the proposed method.
3. The paper shows that the proposed method has better empirical performance than many noisy learning baselines.
4. The proposed method can be combined with existing baseline methods to achieve better performance.

Questions and Weaknesses:
1. I am not very clear about the intuition behind the proposed metric. If the predicted class and the labeled class have almost equal confidence (but they are different classes), e.g., 0.5 and 0.5, then such a sample is always selected as a clean sample but the given label is used for training. Why is this a valid decision?
2. For the CONFES algorithm, it starts with a higher threshold, which selects more data points and gradually decreases the threshold. In such a case, how many data points do the algorithm typically selects in the end? Could it be the case that many samples never get selected and trained?
3. Suppose a noisy-labeled sample by chance have the predicted class to be the same as the given class, such a sample is selected and it seems likely that such a sample will remain to predict the given label and selected in every epoch. Do you observe this happening in practice and if it is the case, how to prevent it?
4. Can you give more justifications for the duplication step of CONFES? Does duplication give better performance and why?

Minor:
1. Page 3: two right parentheses in "The k-class/label classifier F(xi; θ))".

---

> ### Author Response · Authors · 2023-07-11
> **Response to Reviewer zMtf**
>
> We thank the reviewer for the helpful comments and kindly invite the reviewer to check out the revised draft, too (the changes are colored in blue).
>
> > 1. I am not very clear about the intuition ... . Why is this a valid decision?
>
> The predicted label corresponds to the label on which the model has the highest confidence. When the model’s confidence in two distinctive labels is high and close, it means that the model is unsure about the correct label and considers both possibilities to be equally probable. In such cases, our proposed metric takes a conservative approach and considers this sample as a possibly clean sample so that this sample has the chance to be included in the training. In practice, we observe that after the initial training epochs, it is rare for the model to be highly confident in two classes simultaneously. Instead, it tends to become more confident in one class rather than having high confidence in multiple classes.
>
> > 2. For the CONFES ... . how many data points do the algorithm typically selects in the end? Could it be the case that many samples never get selected and trained?
>
> This is not an issue that arises in practice. We updated the manuscript and included Figure 8, which visualizes the number of clean samples that CONFES selects correctly for different noise types and noise rates. In the figure, the distance between the solid and dashed blue lines represents the number of clean samples that CONFES fails to detect, which is very small. The results demonstrate that CONFES effectively distinguishes between clean and noisy samples, with a minimal number of missed identifications of truly clean or noisy samples.
>
> All the clean samples are either selected correctly or rejected incorrectly by CONFES. At the end of the training, the number of truly clean and falsely noisy samples when training PreActResNet18 model on CIFAR-100 in different noise settings is as follows:
>
> ||instance-40%|symmetric-40%|parflip-30%|
> |----------|----------|----------|----------|
> |false noisy rate|6.9%|6.9%|7.6%|
> |# of falsely noisy|2078|2077|2654|
> |true clean rate|93.1%|93.1%|92.4%|
> |# of truly clean|27799|28009|32191|
>
> > 3. Suppose a noisy-labeled sample ... . Do you observe this happening in practice and if it is the case, how to prevent it?
>
> When the model's prediction matches the given label for a sample, but the given label is noisy, it indicates that the model has made an incorrect prediction. As the training proceeds, that specific sample may or may not be selected again depending on the new predicted label. However, at each training epoch, there exist such samples, which are falsely clean samples. In Figure 8, the distance between the solid and dashed orange lines represents the number of samples that have been selected incorrectly. In practice, the number of such samples is small because CONFES excludes the possibly noisy samples from the first training epochs when the model has not started the memorization phase. This is justified by previous research, indicating that generalization occurs in the initial epochs of training while memorization gradually unfolds afterward [1, 2]. At the end of the training, the number of these samples in our experiments on CIFAR-100 with different noise settings are as follows:
> ||instance-40%|symmetric-40%|parflip-30%|
> |----------|----------|----------|----------|
> |% of falsely clean|4.6%|3.1%|8.1%|
> |# of falsely clean|935|902|1228|
>
> We can add the relabeling mechanism, especially in the early epochs, before those noisy samples mislead the model to alleviate this problem. We mentioned it as an interesting direction for future work in the revised draft.
>
> [1] Stephenson, Cory, et al. "On the geometry of generalization and memorization in deep neural networks." ICLR (2021).
>
> [2] Liu, Sheng, et al. "Early-learning regularization prevents memorization of noisy labels." NeurIPS (2020)
>
> > 4. Can you give more ... . Does duplication give better performance and why?
>
> Yes, duplicating the clean samples improves the performance. For instance, when training a PreActResNet18 model on CIFAR-100 with instance-dependent noise of level 40%, the duplication leads to a noteworthy 5.8% increase in test accuracy.
> Sieving the clean samples results in a reduction in the number of training samples due to the exclusion of noisy samples. To account for this reduction and emphasize learning the clean samples, we duplicate the clean samples to maintain the original size of the training set, ensuring it consists only of potentially clean samples. Moreover, in line with a  previous study [3], which indicates duplication is a very strong promoter of learning, duplicating clean samples produces a very strong learning signal which improves the algorithm overall. We clarified this in the last paragraph of section 3.4 in the revised draft.
>
> [3] Carlini, Nicholas, et al. "Quantifying memorization across neural language models." ICLR (2023).
>
> > 5. Page 3: two right parentheses ...".
>
> We fixed it.

---

### Review · Reviewer_zGci · 2023-07-26

**Summary Of Contributions:**


This paper presents a new metric aimed at discerning between noisy and clean labels. The theoretical foundation of the metric is also established, demonstrating its effectiveness in expectation by showcasing a low error rate for the identified labels. Furthermore, empirical evidence is provided to validate the metric's efficacy. Additionally, the paper proposes a method that strategically filters out noisy examples early in the learning process. The experimental results serve to substantiate the efficiency of this proposed approach and its ability to improve overall learning performance by leveraging the metric to handle noisy data.

**Audience:**

Yes

**Claims And Evidence:**

Yes

**Requested Changes:**

1. Theoretically, the paper's proposed method is closely related to the LRT [Zheng et al. 2020]. The paper even uses part of their theoretical results to form one's own. So except for the empirical studies, what is the relationship between the current paper and LRT? Section 2 needs more discussion on this part.
2. Any empirical results demonstrate the efficiency of the proposed method compared to those two-model methods? I understand a trade-off is possible if the proposed method is faster but less accurate. However, if the proposed method is quicker and more accurate, I am wondering why this happen.
3. The clarity of presenting the theoretical results needs to be improved: please define all notations clearly before presenting the concrete results.

**Strengths And Weaknesses:**

Strengths,
1. The paper proposes a novel metric to differentiate clean labeled data and noisy labeled data, which has been demonstrated to be effective at differentiation empirically.
2. An empirical effective method is proposed based on the metric
3. The paper is well-written and well-organized. It is easy to follow.

Weakness
1. Some of the notations lack definition in the theoretical results, such as the \mathfrak{O}
2. The paper's motivation includes that two-model methods are less efficient, and this motivation may imply there is a trade-off between efficiency and accuracy. Such kind of results have not been presented in the paper.

---

> ### Author Response · Authors · 2023-08-07
> **Response to Reviewer zGci**
>
> We thank the reviewer for the helpful comments and kindly invite the reviewer to check out the revised draft, too (the changes are colored in blue).
>
> > 1. Theoretically, the paper's proposed method is closely related to the LRT [Zheng et al. 2020]. The paper even uses part of their theoretical results to form one's own. So except for the empirical studies, what is the relationship between the current paper and LRT? Section 2 needs more discussion on this part.
>
> We thank the reviewer and appreciate the opportunity to clarify this point.
> In terms of similarity, both our method and LRT employ confidence-based sample sieving metrics: confidence error and likelihood ratio, respectively. In terms of differences, unlike the likelihood ratio (employed in LRT), which is a relative metric, confidence error (our work) is an absolute metric. Considering that, our metric is less sensitive to the model quality than the likelihood ratio. Thus, our method can be used from the very initial epochs to sieve the samples. On the other hand, LRT requires using all samples, including noisy ones, for warm-up epochs (e.g., 30) to obtain a sufficiently trained model before starting sieving. In the theoretical analysis, this difference is reflected in the sample sieving metric and its optimal threshold (delta).
>
> To show the effectiveness of our metric, we extensively compare it with LRT in the discussion section (CONFES versus LRT, page 13). Our confidence error metric offers two advantages over the likelihood ratio: (1) Early sample sieving during training. (2) Improved efficiency in distinguishing clean and noisy samples. Our empirical results in Figures 9 and 10 (pages 19 and 20) demonstrate that confidence error exhibits a much lower error rate in identifying noisy samples, making it a superior metric compared to the likelihood ratio for distinguishing between the clean and noisy distributions.
> We expanded section 2 and discussed this part further.
>
> > 2.  Any empirical results demonstrate the efficiency of the proposed method compared to those two-model methods? I understand a trade-off is possible if the proposed method is faster but less accurate. However, if the proposed method is quicker and more accurate, I am wondering why this happen.
>
> As shown in Table 3, our method outperforms Co-teaching, which is based on the small-loss trick and employs two models. Table 4 highlights the efficiency of our metric (confidence error) when it is used as the sieving metric instead of loss value in two-model methods, including Co-teaching and JoCoR. According to Table 4 results, integrating our metric into two-model methods enhances accuracy by 2-5%, albeit at the expense of higher computational time due to the utilization of two models
>
> The reviewer is correct on the trade-off between computational cost and utility, which we also found in our study. The subtle point is that this trade-off holds when employing the same sieving metric. Concretely, when employing the same sample sieving metric (e.g., confidence error), the two-model methods exhibit higher accuracy but slower computational time. However, this trade-off may not be evident if the sieving metrics differ between one-model and two-model methods.
>
> >  3. The clarity of presenting the theoretical results needs to be improved: please define all notations clearly before presenting the concrete results.
>
> We thank the reviewer for the suggestion. We included Table 1 (on page 4),  which provides a comprehensive summary of all the notations used in the theoretical analysis, along with their respective definitions.

---

### Decision · Action_Editors · 2023-08-31

**Recommendation:** Accept as is

**Comment:**

After a major revision, significant additional evidence for the proposed method has been included.  This has been appreciated by all reviewers, resulting in a unanimous recommendation of acceptance.

**Audience:**

It is agreed by all reviewers and myself that the submission is within scope for the journal.

**Claims And Evidence:**

The reviewers are now convinced that the submission has well matched claims and evidence.  They unanimously recommended acceptance.  The decision recommendation of Reviewer zGci summarizes well the contributions and evidence.